# An Evaluation of the Physicochemical Properties of Stabilized Oil-In-Water Emulsions Using Different Cationic Surfactant Blends for Potential Use in the Cosmetic Industry

**Pamela Agredo [1], Maria C. Rave [1], Juan D. Echeverri [1], Daniela Romero [2] and Constain H. Salamanca [1,2,*]**

[1] Programa de Maestría en Formulación de Productos Químicos y Derivados, Facultad de Ciencias Naturales, Universidad Icesi, Calle 18 No. 122–135, Cali 76003, Colombia; pamelasanin@usc.edu.co (P.A.); mdrave@icesi.edu.co (M.C.R.); jdecheverri@icesi.edu.co (J.D.E.)

[2] Departamento de Ciencias Farmacéuticas, Facultad de Ciencias Naturales, Universidad Icesi, Calle 18 No. 122–135, Cali 76003, Colombia; Daniela.romero@icesi.edu.co

[*] Correspondence: chsalamanca@icesi.edu.co; Tel.: +57-2-5552334

**Abstract:** One of the most complex problems in hair care formulations is the duality of the surfactants used. In this regard, such surfactants must be cationic so as to interact with the negatively charged cuticle surface of hair. However, these interdependencies typically lead to non-ideal values for the required hydrophilic–lipophilic balance (HLB) in the oil phase. This study was designed to evaluate the physicochemical properties of several oil-in-water emulsion prototypes for the potential use in hair conditioners. Here, a base formulation was utilized, incorporating binary mixtures of cationic surfactants in different proportions. The cationic surfactants employed were hydroxyethyl-behenamidopropyl-diammonium chloride, behentrimonium methosulphate, cetrimonium chloride, and (iv) Polyquaterniumpolyquaternium-70. The surfactants were evaluated for their capability to decrease the surface tension in an aqueous solution through contact angle measurements between the oily phase and the aqueous phase. The required HLB of the oil phase was also determined. The emulsification process was developed using standard preparation methods. For three months, the prototypes with high viscosity were packed in containers and stored in a stability chamber at accelerated conditions (40 ± 2 °C and 75 ± 5% RH). During this time, the size, size polydispersity, zeta potential, viscosity, rheological profile, and creaming index were all evaluated monthly. The results showed a slight change in the physical stability of the prototypes, where the droplet size increased moderately, however, did little to destabilize the formulations. This suggests that the mixtures of cationic surfactants used could be useful for technological developments in hair conditioning products.

**Keywords:** cationic surfactant blends; oil-in-water emulsions; hair conditioner; stability

## 1. Introduction

A large part of formulating hair conditioning products corresponds to improving oil-in-water emulsion systems that are characterized primarily by having a vast array of raw materials. Among these, preservatives, organoleptic modifiers, and stabilizers constitute some of the main additives in hair conditioners [1]. Such ingredients are usually complex mixtures made up of neutral and cationic surfactants (both monomeric and polymeric) that are used for various purposes in the formulations [2–6]. In the case of neutral surfactants, these are employed with the aim to lower

the interfacial tension between the oil and water, in order to generate a compact film around the droplets in the dispersed phase [4]. Cationic surfactants are utilized for various purposes, particularly for the provision of an electrostatic stabilizing effect, which prevents the aggregation and coalescence of the oil phase [7,8], and to confer the effects of adhesion on hair fibers (especially the cuticle) [9]. Although cationic surfactants are fundamental to formulations in hair care products, at times they lead to unfavorable thermodynamic conditions, as ionic functional groups (i.e., tertiary amines and ammonium salts) provide a highly hydrophilic tone to the required hydrophilic–lipophilic balance (HLB) of stabilizing system mixtures [10–12]. Such a situation leads to differences between the required HLB of the oily phase and the HLB resulting from each of the surfactant mixtures used in the formulation [10,13]. Therefore, to further improve the stability of these emulsified products, it has been necessary to use viscosifying agents that generate a structured vehicle robust network structure [14–16], that avoids aggregation and, in turn, generates important organoleptic features needed for these types of products.

Although the first impression of this work could be lacking in novelty about the stabilization of emulsified systems, it should be highlighted that the information available in the specific field of hair care products is minimal and limited. Only a handful of studies on the stabilization of these types of products have been reported in literature, much less those where binary cationic surfactant mixtures are used [1,17,18]. For this reason, we focus this study on evaluating the effects of four cationic surfactants (Figure 1), commonly used as raw materials in the cosmetics field, where their applications have yet to be reported widely, particularly the synergistic effects on the stabilization of hair conditioner products.

**Figure 1.** Chemical structures of cationic surfactants used in the study.

## 2. Materials and Methods

### 2.1. Materials

Hydroxyethyl-behenamidopropyl-dimonium chloride (HBD-C1) and behentrimonium-methosulfate were provided by CRODA (Snaith, United Kingdom), whereas cetrimonium chloride was supplied by BASF (Ludwigshafen, Germany). The remaining raw materials were provided by the cosmetic company Belleza Express S.A. (Cali, Colombia) and used as received. Water Type II (ultra-pure water) was obtained from a purification system called Millipore Elix Essential (Merck KGaA, Darmstadt, Germany).

### 2.2. Surface Tension and Contact Angle Measurements

Surface tension and static contact angle measurements were carried out using the pendant and sessile drop methodology [19–23]. Here, a video-based optical contact angle instrument (OCA15EC Dataphysics Instruments, Filderstadt, Germany) with version 4.5.14 SCA20 and SCA22 software were used. Data were recorded on an IDS video camera, where information was gathered from approximately 400 to 800 frames for reference as the static angle. Moreover, the point of capture was defined where the reflected incident light completely disappeared (about 1 s from leaving the dispensing system). Drop volumes ranged from $5 \times 10^{-3}$ to $15 \times 10^{-3}$ mL, whereas the liquid deposition was fixed to 1 cm for all assays. Each measurement was carried out at approximately $22 \pm 1$ °C and $60\% \pm 5\%$ relative humidity.

### 2.3. Determination of the Required HLB for the Oil Phase

To determine the required HLB in the oil phase for the emulsified prototypes, 11 formulations were synthesized (in the oil phase only) using a range of preservatives, ultra-pure water, and an emulsifying system. In these formulations, the same proportions were maintained as described in Table 1. Only the emulsifying system was changed to a Span 80 and Tween 80 blend, with the mixture combined using different proportions to obtain mixed HLB values, corresponding to 6, 7, 8, 9, 10, 11, 12, 13, 14, 15, and 16. Subsequently, each prototype was packed inside a 15 mL Falcon™ tube and exposed to a thermal stress assay for three weeks. The storage conditions were changed from $40 \pm 1$ °C to $4 \pm 0.5$ °C every four days.

**Table 1.** Hair conditioner product prototype formulations.

| Individual Component | Ingredients | % (w/w) |
|---|---|---|
| Oil Phase | Cetearyl-alcohol Coco-caprylate Shea butter | 4.8 |
| Viscosity agent | Hydroxy-ethyl-cellulose | 0.5 |
| Wetting agent | Glycerin | 0.2 |
| Preservative | Methyl-isothiazolinone Phenethyl-alcohol Propylene-Glycol-2-methyl ether | 0.2 |
| pH modifier | Citric acid | 0.05 |
| | Lauryl-glucoside (neutral) | 0.2 |
| Emulsifier system | Cationic Surfactant 1 Cationic Surfactant 2 | 1 |
| Dispersing phase | Water | q.s. |

### 2.4. Preparation of Emulsions

The emulsified systems were developed according to the formulation shown in Table 1.

All prototypes were synthesized in triplicate using different binary cationic surfactant blends, as shown in Table 2. At first, the aqueous phases (composed of hydroxy-ethyl-cellulose, glycerin, and ultra-pure water) were mixed using a homo-mixer (Ultra Turrax® T-25, IKA®, Staufen, DE-BW, Germany) at 6000 rpm for 10 min and subsequently heated to 80 °C (mixture 1). At the same time, the oily ingredients were weighed together with the surfactants (mixture 2) and heated to 75 °C in a separate vessel. The oil phase (mixture 2) was then added to the aqueous phase (mixture 1) with a 9000-rpm agitation for 10 min until a white emulsion was formed with a viscous consistency. The emulsions were continuously stirred at 400 rpm speed using a propeller-type stirrer (IKA® RW 20, IKA®, Staufen, DE-BW, Germany) and heated to 40 °C, whereas the remaining ingredients (i.e., preservative and pH modifier) were incorporated into the emulsion. The resulting mixture was cooled to room temperature. Subsequently, the prototypes were selected according to the viscosity

criteria, which must have a minimum viscosity of 15,000 cP. These conditions are fundamental to both intrinsic organoleptic characteristics and the stabilization of these products.

**Table 2.** Designing the cationic surfactant mixture for the hair conditioner prototypes.

| Emulsified System Prototype | Cationic Surfactant Mixture | Ratio |
| --- | --- | --- |
| 1 | HBD-Cl and BT-MS | 1:3 |
| 2 | HBD-Cl and BT-MS | 1:1 |
| 3 | HBD-Cl and BT-MS | 3:1 |
| 4 | HBD-Cl and CT-Cl | 1:3 |
| 5 | HBD-Cl and CT-Cl | 1:1 |
| 6 | HBD-Cl and CT-Cl | 3:1 |
| 7 | HBD-Cl and PQ-70 | 1:3 |
| 8 | HBD-Cl and PQ-70 | 1:1 |
| 9 | HBD-Cl and PQ-70 | 3:1 |
| 10 | BT-MS and CT-Cl | 1:3 |
| 11 | BT-MS and CT-Cl | 1:1 |
| 12 | BT-MS and CT-Cl | 3:1 |
| 13 | BT-MS and PQ-70 | 1:3 |
| 14 | BT-MS and PQ-70 | 1:1 |
| 15 | BT-MS and PQ-70 | 3:1 |
| 16 | CT-Cl and PQ-70 | 1:3 |
| 17 | CT-Cl and PQ70 | 1:1 |
| 18 | CT-Cl and PQ-70 | 3:1 |

The ratio is given in parts that are 1% of the total cationic surfactant in each formulation. HBD-Cl: Hydroxyethyl-behenamidopropyl-diammonium chloride, BT-MS: Behentrimonium methosulphate, CT-Cl: Cetrimonium chloride, and PQ-70: Polyquaternium-70.

### 2.5. Accelerated Stability Tests

One hundred milliliters of each prototype conditioner was packed in a polyethylene-terephthalate-PET container and stored at accelerated stability conditions (40 °C $\pm$ 2 °C and 75 $\pm$ 5% relative humidity) for 12 weeks.

### 2.6. Zeta Potential, pH, and Conductivity Measurements

Zeta potential measurements were carried out using a zetasizer nano ZSP (Malvern Instruments, UK) at 25 $\pm$ 2 °C temperature, with equilibration times of 120 s in a DTS 1070 capillary cell. Here, the attenuator position and intensity were set automatically. To prepare a sample, 130 mg of the emulsifier was diluted in 20 mL of ultra-pure water and manually stirred. From this, a 50 µL aliquot was taken and diluted with 1 mL of ultra-pure water before each zeta potential measurement was taken. The electrical conductivity and the pH of the emulsions were determined using a CR-30 conductivity meter and a Starter-2100 pH meter, respectively.

### 2.7. Particle Size Analysis

The particle size distribution of the emulsions was obtained using a Mastersizer 3000 (Malvern Instruments, Malvern Instruments, UK), equipped with a helium/neon laser at a wavelength of 632.8 nm. Previously, 0.6 g of the emulsion was diluted with 10 mL of ultra-pure water at 25 $\pm$ 2 °C and stirred at 400 rpm. The appropriate amount sample was determined when the dark level was reached (between 2% and 8%).

### 2.8. Rheological Profile and Viscosity

The rheological profile was obtained using different shear rates ($20 \text{ s}^{-1}$ to $150 \text{ s}^{-1}$). For this, a viscometer (micro-visc, RheoSense Inc., San Ramon, CA, USA) was used. The viscosity was measured using a Brookfield® viscometer with a No. 4 needle (Brookfield®, Middleboro, MA, USA) at 10 rpm.

*2.9. Creaming Index (CI)*

Creaming index values were determined from the ratio of sediment volume and the total emulsion volume. Here, 15 mL of the emulsion was centrifuged in a Wincom 80-2 centrifuge at 3000 rpm for 4 h. The results were expressed as the creaming index (CI) [24,25] according to

$$CI = \frac{H_S}{H_E} \times 100 \tag{1}$$

where $H_S$ is the sediment height and $H_E$ is the sample height before centrifugation.

## 3. Results and Discussion

One of the most important considerations in the preformulate-on stages of emulsified systems is the adequate selection of those surfactants that can lead to the maximum stabilization of the heterodisperse system. According to this, there are several aspects that should be considered at the time of such selection, where (i) the capability to decrease surface tension and (ii) the capability to provide a surfactant mixture's HLB value very close to the required HLB value of the oily phase, are the most important considerations. Due to the aforementioned, the first approach of the study was to evaluate the surfactant's capability to decrease the surface tension of water, as well as to determinate the required HLB value of the oil phase in the formulation.

*3.1. Surface Tension and Contact Angle Measurements*

The resulting reduction in surface tension and contact angle given by the surfactants are shown in Figure 2.

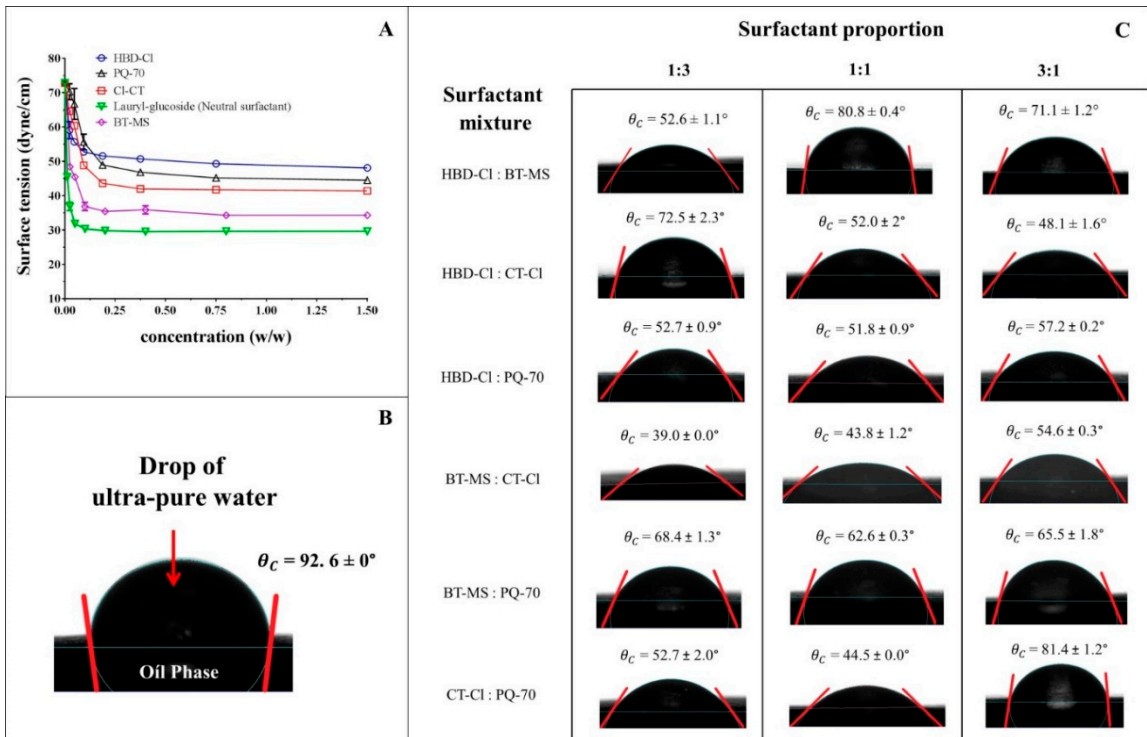

**Figure 2.** (**A**) The reduced surface tension due to the concentration of surfactants used in the hair conditioner formulation. (**B**) A drop of ultra-pure water lying on top of the oil phase. (**C**) Drops of the cationic surfactant blends in aqueous solution lying on top of the oil phase. HBD-Cl: Hydroxyethyl-behenamidopropyl-diammonium chloride, BT-MS: Behentrimonium methosulphate, CT-Cl: Cetrimonium chloride, and PQ-70: Polyquaternium-70.

Figure 2A shows that the surfactant amounts used in the prototype formulations are slightly higher than the critical micellization concentration (cmc) values for each received surfactant in aqueous media. This could be because the emulsifiers in the right quantities were needed to reach the stabilization point in the heterodisperse system. In addition, it was observed that the neutral surfactant (lauryl-glucoside) proved to be the most effective in reducing surface tension, which is in agreement with that previously reported in the literature [26–28]. Likewise, such neutral surfactant has an HLB value of 10 according to the Griffin method (see supplementary file), which makes it in the formulation´s surfactant with the HLB value closer to the 'idealized zone' for the stabilization of oil-in-water emulsions (HLB = 8–16) [29]. Conversely, the surfactants Cl-CT and HBD-Cl have HLB values of 22 and 32, respectively, which are outside the ideal range to stabilize oil-in-water emulsions. In the case of PQ-70, this surfactant has an HLB value of 13, and although it is in the ideal zone for stabilization, its capability to decrease the surface tension is very low, making the surfactant less effective [29].

In contrast, Figure 2B indicates that the boundary generated between the water and the oil phase has a hydrophobic interface of $\theta c$ = 92.6°, [20,21,23], which decreases when the surfactants were incorporated into the aqueous phase (Figure 2C). Such a marked decrease was more evident in the mixtures that contained the CT-Cl surfactant, as opposed to the mixtures containing the HBD-Cl or PQ-70 surfactants. These results prove very interesting, considering that the largest proportion of the oil phase is made up of cetearyl alcohol and coconut caprylate, which are fatty compounds with alkyl chains composed of 14–16 carbon atoms. It seems the cetrimonium chloride (CT-Cl) surfactant, which has an alkyl chain length of 14 carbon atoms, is the emulsifier with a higher affinity to the oil phase.

### 3.2. Determining the Required HLB for the Oil Phase

According to the thermal stress study conducted, it was found that the lowest creaming index values were obtained when the system presented a surfactant HLB value between 12 and 13 (Figure 3).

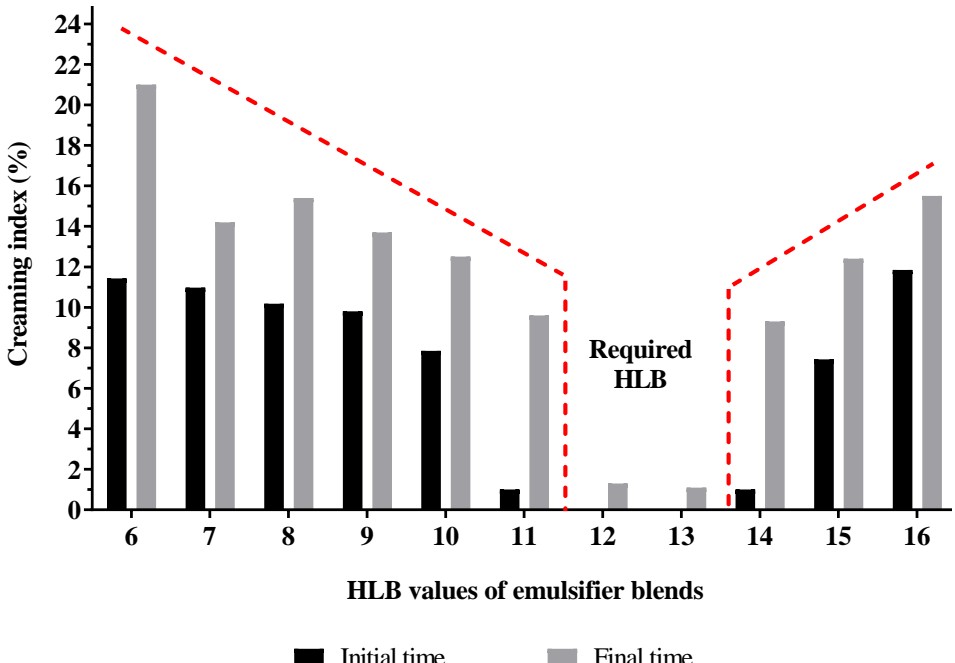

**Figure 3.** Results of the creaming index with respect to the hydrophilic–lipophilic balance (HLB) values of the cationic surfactant blends used for the emulsified prototypes.

The results of the required HLB were consistent in the oil phase formulations, which were mainly composed of fatty components having an intermediate alkyl chain length containing 14 and 16 methylene groups and also alcohol-like functional groups, which decreased the hydrophobicity. This result is consistent because it matches with HLB value of 12.4 obtained by the Griffin method [11]

(See supplementary file). Therefore, the values of the creaming index suggest that in order to achieve the best physicochemical properties with stabilized emulsions, it is necessary to use a combination of surfactants with an HLB close to the required HLB of the oil phase.

### 3.3. Emulsion Preparation

The results of the viscosity and mixed emulsifier HLB used in the hair conditioner prototypes are shown in Figure 4. Also, the calculations used to determine the HLB values for the emulsified prototypes are detailed in the supplementary material. In regard to the pH of the emulsified systems, these were adjusted between 4.0 and 4.5. This is due to the required pH values of these types of products [30].

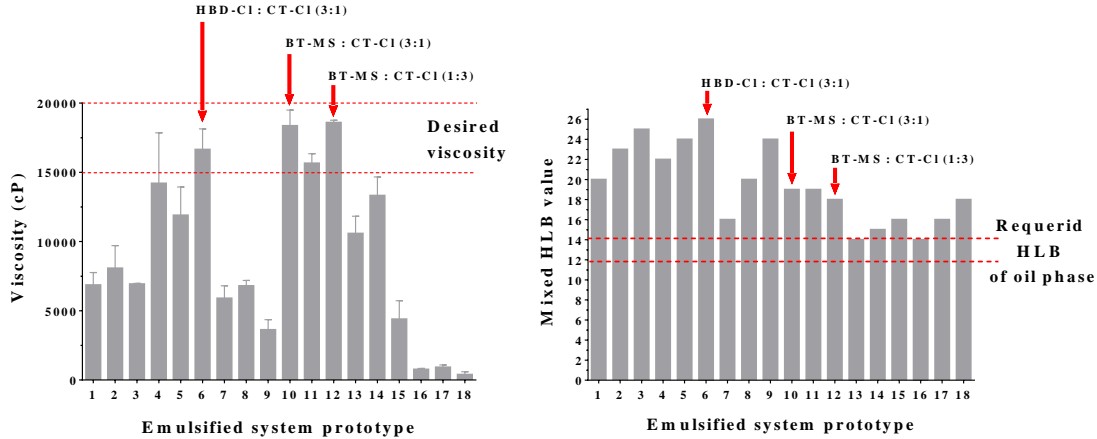

**Figure 4.** Results of viscosity and mixed HLB values for the emulsified prototypes.

These results show that the cationic surfactant mixtures needed to obtain the required HLB for the oil phase (i.e., emulsified system prototypes 13 and 16) did not reach the viscosities needed for these products [16]. The cationic surfactant blends that did achieve the required viscosity, however, had HLB values greater than those required in the oil phase. This result indicates that it was not possible to find a cationic surfactant mixture system that would achieve the HLB required by the oil phase, and the viscosity required for this class of product. According to the previous result, the emulsions that fulfilled the viscosity requirement were selected, even if they were not in the "ideal condition" of the required HLB [31,32]. The prototypes selected were the system-6 [HBDC-Cl:CT-Cl (3:1)], the system-10 [BT-MS:CT-Cl (1:3)], and the system-12 [(BT-MS:CT-Cl (3:1)].

### 3.4. Accelerated Stability Tests

For the selected prototypes, the results of the creaming index were unable to show a phase separation. The formation of the layer that suggested "the emulsion break" or the separation of phases was not observed and thus, the sedimentation rate could not be determined, or was assumed with a value of 0 (zero). This suggests that the selected systems do have adequate stability characteristics. Nevertheless, in order to obtain a more detailed description of the stabilization process, it was necessary to analyze other parameters, such as the droplet size, pH, electrical conductivity, zeta potential, and rheological profile. All these results are shown and explained in detail in the following Sections 3.5 and 3.6.

### 3.5. pH, Electrical Conductivity, Zeta Potential, and Droplet Size

The results of the pH, electrical conductivity, zeta potential, and oil droplet size for the chosen prototypes are shown in Figure 5.

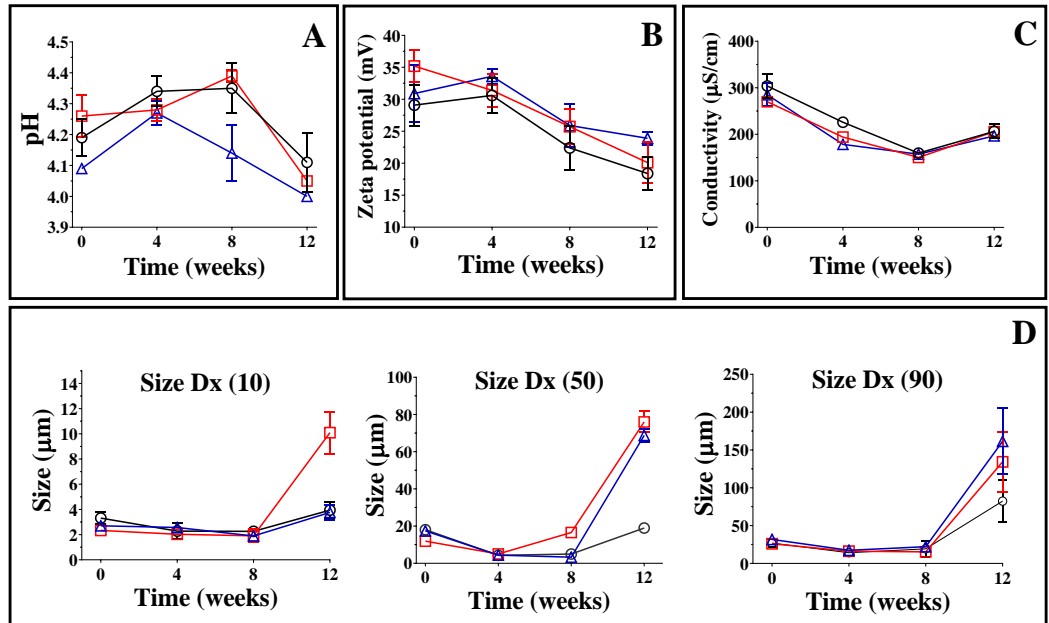

**Figure 5.** Results of (**A**) pH, (**B**) electrical conductivity, (**C**) zeta potential, and (**D**) particle size distribution for the selected prototypes as a function of time. ○ HBD-Cl + CT-Cl (3:1), □ BT-MS + CT-Cl (1:3), △ BT-MS + CT-Cl (3:1).

It can be seen in Figure 5 that the selected prototypes displayed similar behaviors in most of the cases evaluated. For the pH measurements (Figure 5A), no significant changes were observed during the course of the 12-week study around pH values between 4 and 4.5. Conversely, the zeta potential presented a decrease with respect to time for the selected prototypes (Figure 5B). In the first few weeks, the zeta potential changed from ~+35 mV to ~+30 mV for the surfactant blend BT-MS + CT-Cl (1:3). Conversely, blends HBD-Cl + CT-Cl and BT-MS + CT-Cl (3:1) remained constant. Here, the zeta potential was found to decrease to +20 and +25 mV in the very last weeks for these three prototypes. This behavior could be ascribed to the migration of cationic surfactants from the oil–water interface to the dispersion medium [33]. In this regard, when the cationic surfactants migrated toward the dispersing bulk phase, the emulsion interface lost part of its electrostatic polarization capabilities, which caused the zeta potential to decrease. This then caused the electrical conductivity to increase as the surfactant counterions were less attracted to the double electric layer formed by the oil–water interface. It then gained better electrical mobility and thus greater conductivity (Figure 5C) [34,35]. Regarding to droplet size (Figure 5D), the values obtained for DX (90) contained very large polydisperse droplet sizes, which may have been formed because of the incorporation of air during the manufacturing process. Instead, the values obtained at the 4th and 8th weeks showed that both the size and polydispersity lowered, probably due to the loss of air previously incorporated into the emulsified preparation. At the 12th week, however, it was found that the size of the droplets increased significantly, suggesting that this was the point at which the aggregation processes began [36]. The values obtained for DX (10) and DX (50) showed no variation at 4th and 8th weeks, meaning that the particle size remained stable with no aggregation experienced. The systems did, however, start to aggregate at week 12. An example is the emulsion formed by HBD-Cl:CT-Cl (3:1), which showed fewer variations in size, suggesting that a surfactant mixture of this type was one that projected the highest stability.

### 3.6. Rheological Profile and Viscosity

The results of the rheological behavior for each of the prototypes are shown in Figure 6.

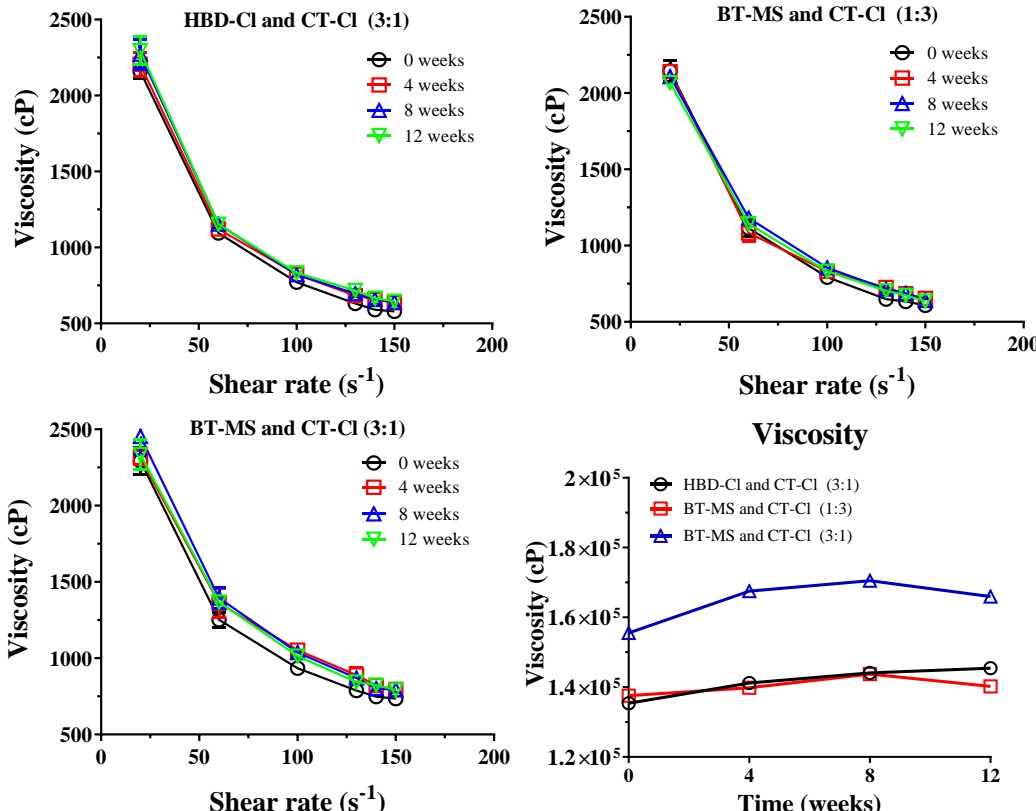

**Figure 6.** Rheological profiles and viscosity values obtained for the selected prototypes as a function of time.

Here, three prototypes were observed to behave in a pseudo-plastic way [6] since the viscosity of the system decreased with an increase in shear rate. Such rheological behavior, which was required for these types of products was due to the dosage and application form [14] that remained during the 12 week evaluation process. It is very important that this rheological behavior remained in time, because this ensures that the product (hair conditioner) can be used by consumers correctly [30,37]. In contrast, an increase in viscosity was obtained up until the 8th week, which saw to the viscosity stabilize then decrease at the 12th week. This phenomenon may be attributed to how the emulsions were manufactured; where high shear forces are typically applied, which tends to incorporate air into the system, affecting the cohesiveness of the phases. Over time, however, the systems began to recover their internal configuration [38] and this stability could also be attributed to the Hydroxy-ethyl-cellulose polymer used as the viscosity agent [39]. Nevertheless, the two systems that contain the BT-MS surfactant do start to decrease at week 12, in contrast to the system with the HBD-Cl that remains stable.

## 4. Conclusions

It was found that the required HLB for the oil phase used was between 12 and 13. However, the cationic surfactant blends that most approached this value were incapable of reaching the viscosity values required for these types of products. Therefore, we conclude that our best surfactant blend cannot be explained by the HLB theory because of its cationic nature. Likewise, it was found that the physicochemical characterization and interfacial stabilization was provided instead by the formation of a structured network. This includes the non-ionic surfactant and the Hydroxy-ethyl-cellulose polymer used as the viscosity agent. In regard to the prototypes, they were found to have the appropriate physical stability according to creaming index testing. The pH, electrical conductivity, and zeta potential all indicated that the emulsified prototypes tended to change slightly because of the migration of cationic surfactants from the interface to the dispersing phase. The rheological study showed that

the three emulsified systems displayed pseudo-plastic behavior, which was ideal for these types of products. Furthermore, it was found that such systems change their viscosity over time, due, in part, to the incorporation of both air and high strength shears in the initial manufacturing process, along with the subsequent structural rearrangements in the emulsion. The particle size results showed a decrease in kinetic stability for these systems with a mixture of emulsifiers containing BT-MS:CT-Cl (3:1) and BT-MS:CT-Cl (1:3), due to the aggregation of the internal phase. The system that contained a surfactant blend with HBD-Cl:CT (3:1), however, did remain stable, which suggests that this combination is useful and may be an interesting alternative to formulations used for hair conditioners.

**Supplementary Materials:** The following are available online at http://www.mdpi.com/2079-9284/6/1/12/s1, 1. HLB calculated of Surfactants, 2. Required HLB of oil phase.

**Author Contributions:** P.A. made and analyzed the stability assay (from emulsion preparation to rheological profile and viscosity), M.C.R. and J.D.E. made the surface and interfacial characterization, whereas D.R. did the determination of required HLB. C.H.S. designed the experiments, analyzed, and discussed the data and wrote the manuscript.

**Funding:** This research received no external funding.

**Acknowledgments:** The authors thank Icesi University for providing the funding for the execution of this research work and Belleza Express S.A. from Colombia for providing the raw materials.

**Conflicts of Interest:** The authors declare no commercial interest that could represent a conflict related to the study.

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
