# Peer review of "An Evaluation of the Physicochemical Properties of Stabilized Oil-In-Water Emulsions Using Different Cationic Surfactant Blends for Potential Use in the Cosmetic Industry"

_cosmetics, doi:10.3390/cosmetics6010012_

Round 1
Reviewer 1 Report
1) authors continuously make reference to the HLB method for a proper emulsions design. Indeed, the HLB concept is today largely overseeded, because it cannot be applied indiscriminately to all emulsions. Moreover, as this same study demonstrates, the HLB application is not sufficient to forecast and establish the stability of the resulting emulsions. Dimensional, geometrical, solubility, structural and additional stabilization factors have far higher rimportance. In other words, it is completely unseful to search for the theoretical compliance with the HLB numbers, especially when they cannot be matched by the used emulsifiers. As authors have collected a sufficient number of other physical parameters and evaluations, the HLB concept could be mentioned only to evidence the difference between its predictions and the practical experimental work.
This study would be better described as development strategy of emulsions where the HLB concept cannot be applied.
2) It is a pity that viscosity values have been measured on samples that have not been well deareated. Particularly considering that the used emulsifiers have a good potential of stabilizing entrapped air bubbles. This factor has a deep (and unknown) influence on stability of the samples and time-evolution of the dimensions on the dispersed phase. And, consequently, of conductivity measures. Measures shoul dbe made on deareated samples
3) Just a suggestion for getting better stability and improved particle size is to add slowly 20% of the hot water phase to the oil+emulsifiers blend and then homogenize. The reamining water is added as cold water. This method is particularly useful when the amount of oil phase is relatively small.
Author Response
the answers are in the attached file

Reviewer 2 Report
This manuscript describes an interesting study concerning the assessment of the properties of stabilized Oil-In-Water emulsions using different cationic surfactant. A base formulation was utilized, incorporating binary mixtures of cationic surfactants in different proportions. The surfactants were evaluated for their capability to decrease the surface tension in aqueous solution through contact angle measurements between the oily phase and aqueous phase. The work is interesting; however, there are a few comments which need to be addressed by the authors:
1) The results of the Accelerated stability tests should be reported and presented in more detail, at least as supplementary materials.
2) The discussion of the results in most cases are very poor. The authors should discuss their findings in light of the available literature.
3) The authors in the conclusion section should point out what is the novelty of their study, compared to the available literature. The finding that the surfactant chloride of cetrimonium was the emulsifier with the highest affinity to the oil phase is not enough to justify the merit and the value of the present work.
Author Response
the answers are in the attached file

Reviewer 3 Report
The paper needs some revisions:
1. Page 1, line 42: “In the caseof neutral surfactants…” There is a typing error. “In the case of neutral surfactants…” is correct.
2. Page 2, line 50: “…system mixture mixtures [10-12].” “mixture mixtures” sounds strange.
3. Page 2, line 58: there is an extra gap before the sentence starting “Only a handful…
4. All Figures and tables mentioned in the text are in bold; only Figure 1 (page 2, line 61) is not in bold.
5. Generally, through the body of the paper: If there is a citation in the sentences, sometimes the sentence ends with a full stop after the citation, sometimes a full stop is before the citation. E.g. “…in the dispersed phase [4].” and “…sessile drop methodology.[19-23] The same style should be used.
6. Chapter 2.5.: Rotation speed unit is “rpm”, not “rmp” as used wrongly twice (lines 99 and 104). Viscosity expressed in centipoise – “Cp” is not correct; “cP” is correct. In Figures 4 and 6 the correct “cP” is used.
7. Figure 2 and the text referring to it: There are 3 sub-figures, A, B and C. In the legend and in the text the authors use “a, b and c”. It should be the same.
8. Figure 5 and the text referring to it: the similar comment as to Figure 2. Furthermore, the name of x-axes should start with the capital: Time (weeks).
9. Figure 3, in the legend: “Initian time” is not correct; “Initial time” is correct. The font size seems to be too big as compared to other Figures.
10. Figure 6 (viscosity part): and the text referring to it: the name of x-axes should start with the capital and the unit should be in plural: Time (weeks).
Author Response
the answers are in the attached file

Round 2
Reviewer 2 Report
The authors revise the manuscript to address all comments.